# ARMMs as a versatile platform for intracellular delivery of macromolecules

Qiyu Wang[1], Jiujiu Yu[1,2], Tatenda Kadungure[3], Joseph Beyene[1], Hong Zhang[3] & Quan Lu[1]

Majority of disease-modifying therapeutic targets are restricted to the intracellular space and are therefore not druggable using existing biologic modalities. The ability to efficiently deliver macromolecules inside target cells or tissues would greatly expand the current landscape of therapeutic targets for future generations of biologic drugs, but remains challenging. Here we report the use of extracellular vesicles, known as arrestin domain containing protein 1 [ARRDC1]-mediated microvesicles (ARMMs), for packaging and intracellular delivery of a myriad of macromolecules, including the tumor suppressor p53 protein, RNAs, and the genome-editing CRISPR-Cas9/guide RNA complex. We demonstrate selective recruitment of these macromolecules into ARMMs. When delivered intracellularly via ARMMs, these macromolecules are biologically active in recipient cells. P53 delivered via ARMMs induces DNA damage-dependent apoptosis in multiple tissues in mice. Together, our results provide proof-of-principle demonstration that ARMMs represent a highly versatile platform for packaging and intracellular delivery of therapeutic macromolecules.

[1] Program in Molecular and Integrative Physiological Sciences, Departments of Environmental Health, and Genetics & Complex Diseases, Harvard T.H. Chan School of Public Health, Boston, MA 02115, USA. [2] Department of Nutrition and Health Sciences, University of Nebraska Lincoln, Lincoln, NE 68583, USA. [3] Department of Pediatrics, Division of Genes and Development, University of Massachusetts Medical School, Worcester, MA 01655, USA. These authors contributed equally: Qiyu Wang, Jiujiu Yu. Correspondence and requests for materials should be addressed to Q.L. (email: qlu@hsph.harvard.edu)

Mammalian cells secrete into extracellular milieu a variety of tiny membrane-encapsulated vesicles[1]. These extracellular vesicles (EVs) contain functional molecules such as proteins and RNAs, which can be taken up by recipient cells to mediate intercellular communication[2, 3]. Because of their ability to carry and transfer bioactive molecules, EVs have been proposed as a new vehicle for therapeutic delivery[4, 5]. Since EVs are encapsulated by a lipid bilayer membrane, cargos enclosed in the vesicles are protected from protease- or nuclease-mediated degradation and shielded from possible detection as foreign antigens by the immune system. Several studies have used exosomes, the most common and the best characterized EVs, for delivery of siRNAs[6] and proteins[7]. However, exosome-mediated therapeutic delivery is limited by the inability to control vesicle production and by crude packaging methods (e.g., electroporation). In addition, the biogenesis of exosomes in the degradative late endosomes further reduces the packaging and delivery efficiency[8].

Arrestin domain containing protein 1 [ARRDC1]-mediated microvesicles (ARMMs) are EVs that are distinct from exosomes[9]. The budding of ARMMs requires ARRDC1, which is localized to the cytosolic side of the plasma membrane and, through a tetrapeptide motif (PS/TAP), recruits the ESCRT-I complex protein TSG101 to the cell surface to initiate the outward membrane budding[9]. Thus, in contrast to exosomes, the biogenesis of ARMMs occurs at the plasma membrane. ARMMs exhibit several additional features that make them potentially ideal vehicles for therapeutic delivery. ARRDC1 is not only necessary but also sufficient to drive ARMMs budding. Indeed, simple overexpression of the ARRDC1 protein increases the production of ARMMs in cells[9]. This allows controlled production of ARMMs using modern biological manufacturing methods. Moreover, endogenous proteins such as cell surface receptors are actively recruited into ARMMs and can be delivered into recipient cells to initiate intercellular communication[10], suggesting that the exogenous cargo molecules may be similarly packaged and delivered via ARMMs.

In this study we explore the ability of ARMMs to efficiently package and deliver diverse classes of cargo macromolecules, including p53 protein, RNA molecules, and the CRISPR-Cas9/gRNA complex. In all cases, we show selective packaging of the respective cargos into ARMMs. We demonstrate that the cargo molecules packaged in ARMMs, when transferred to recipient cells or tissues, carry out specific biological functions. Together, our data provide a proof-of-principle demonstration for the utility of ARMMs to package and deliver a variety of bioactive macromolecules.

## Results

**Packaging of p53 protein into ARMMs**. We first tested the ability of ARMMs to package and deliver potential therapeutic proteins. We chose tumor suppressor p53 as the cargo protein, as restoration of p53 protein function in cancer cells triggers apoptosis or senescence[11], leading to regression of tumors with p53 dysfunction[12]. Thus, delivery of functional p53 is a viable therapeutic strategy against many cancers in which p53 function is compromised. During ARMMs biogenesis, ARRDC1 itself is incorporated into the vesicles[9]; thus we reasoned that fusing p53 to ARRDC1 may enable p53 protein to also be incorporated into ARMMs. We made a construct with ARRDC1 fused directly to the N-terminus of wild-type p53 (Fig. 1a). Expression of the ARRDC1-p53 fusion protein (Supplementary Fig. 1a) significantly increased the transcription of p53 target genes, *MDM2*, and *p21*, in H1299 cells (Supplementary Fig. 1b), which lack functional p53 protein[13]. Although less potent than the unfused

wild-type p53 protein in inducing the target gene expression (Supplementary Fig. 1b), ARRDC1-p53 fusion protein retains the important transcriptional activity of p53 protein.

We next determined whether the ARRDC1-p53 fusion protein can be packaged into ARMMs. We first evaluated the vesicle budding activity of the fusion protein. We transfected ARRDC1-p53 into the human embryonic kidney 293T (HEK293T) cells and harvested EVs for assessment by the NanoSight particle analysis. ARRDC1-GFP, which drives ARMMs formation[9], was used as a positive control. As shown in Fig. 1b, the size distribution as well as the amount of EVs secreted from ARRDC1-p53-transfected cells was comparable with those from cells transfected with ARRDC1-GFP. We also used transmission electron microscopy (TEM) to image EVs isolated from cells expressing either control GFP, ARRDC1-GFP, or ARRDC1-p53 (Supplementary Fig. 2). The diameter of most EVs is less than 100 nm, consistent with our previous characterization of ARMMs at 50–80 nm[9]. While not particularly quantitative, the TEM data also showed more EVs from ARRDC1-p53-overexpressing cells than the GFP control. Together these data indicate that the fusion of ARRDC1 to p53 did not compromise the ability of ARRDC1 to drive ARMM generation. Importantly, Western blotting showed that ARRDC1-p53, not endogenous unmodified p53, was present in the EV fractions (Fig. 1c). Interestingly, in the EVs, ARRDC1-p53 as well as ARRDC1-GFP appeared to exist as multiple protein bands (Fig. 1c), which presumably are results of ubiquitination on ARRDC1, as our previous study has shown that ARRDC1 is ubiquitinated and that such ubiquitination enhances ARMMs budding[9]. Together, these data indicate that ARRDC1-p53 functions similarly to ARRDC1 in driving ARMMs budding, and that p53 fusion to ARRDC1 mediates p53 packaging into ARMMs. To quantify cargo packaging in ARMMs, we did Western blotting using recombinant GFP protein standard at a range of 50–1000 ng along with ARMMs produced from ARRDC1-GFP-expressing cells (Supplementary Fig. 3a). From the standardized plot we estimated that each ARMM vesicle contains an average of ~540 cargo protein molecules (Supplementary Fig. 3a).

**Delivery of p53 protein in vitro and in vivo**. To determine whether ARRDC1-p53 in ARMMs can be delivered into recipient cells, we incubated p53-null H1299 cells with either ARRDC1-p53 ARMMs or the control ARRDC1-GFP ARMMs. Western blotting showed that incubation of H1299 cells with ARRDC1-p53 ARMMs led to robust detection of ARRDC1-p53 protein in the recipient cells (Fig. 1d). We used GFP-standardized Western blotting to quantify the amount of ARMMs transfer to recipient cells, and estimated that each recipient cell received an average of $3.1 \times 10^6$ cargo protein molecules from $5.8 \times 10^3$ of ARMMs (Supplementary Fig. 3b). Immunofluorescence staining further showed that ARRDC1-p53 protein delivered via ARMMs was present in the cytoplasm as well as the nuclei of the recipient H1299 cells (Supplementary Fig. 4). Importantly, ARRDC1-p53-containing ARMMs induced transcription of *MDM2* and *p21* in H1299 cells (Fig. 1e), indicating that ARRDC1-p53 protein delivered via ARMMs is able to induce p53-dependent gene expression in recipient cells.

To investigate whether ARMMs can deliver functional p53 in vivo, we used *p53* knockout (KO) mice. In wild-type animals ionizing radiation induces DNA damage, which leads to p53-dependent apoptosis[14]. In *p53* KO mice, DNA damage response and apoptosis are compromised. We tested whether ARMMs can deliver functional p53 protein to restore DNA-damage-induced apoptosis in *p53* KO mice. *P53*-null mice were intravenously injected with ARMMs that contain either ARRDC1-p53 or the

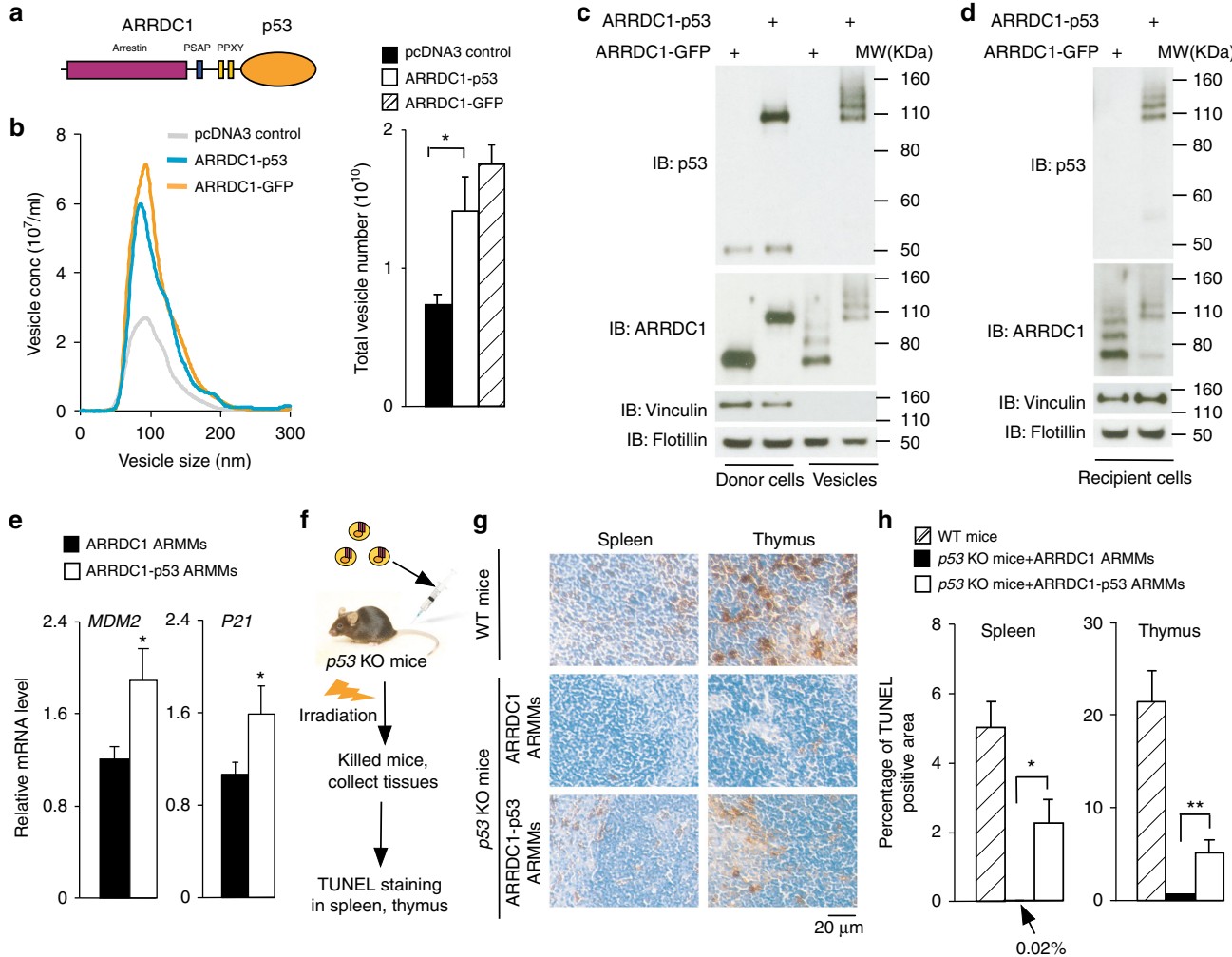

**Fig. 1** Packaging and in vivo delivery of p53 protein via ARMMs. **a** Schematic showing p53 fused to the C-terminus of ARRDC1. Domains and motifs in ARRDC1 relevant to ARMMs budding are indicated. **b** Effect of ARRDC1-p53 expression on the production of EVs. pcDNA3, ARRDC1-GFP, or ARRDC1-p53 construct was transfected in HEK293T cells. EVs in the media were analyzed by the NanoSight NS300 instrument (left graph). Numbers of EVs from 3 independent experiments were quantified (right graph). Data were presented as the mean ± SEM. **c** Packaging of ARRDC1-p53 into ARMMs. pcDNA3, A1-GFP, or A1-p53 was transfected in the production of HEK293T cells. EVs in medium were pelleted vesicles by ultracentrifugation. Western blotting for p53, ARRDC1, and control protein vinculin was done on both cell lysates and EVs. **d** Transfer of ARRDC1-p53 fusion protein in ARMMs to recipient cells. Purified ARMMs containing ARRDC1-GFP or ARRDC1-p53 were incubated with H1299 cells overnight. The cells were washed with PBS extensively and subjected to Western blot analysis. **e** Induction of p53 target genes by ARRDC1-p53 fusion protein delivered via ARMMs. ARRDC1- or ARRDC1-p53 ARMMs were incubated with H1299 cells for 48 h. Total RNAs were extracted from the cells and used for qRT-PCR to measure MDM2 and p21 mRNA expression. Data were presented as the mean ± SD. The experiments were repeated for five times. **f** Schematic of in vivo delivery strategy. Control or ARRDC1-p53 ARMMs were injected in the tail veins of p53 knockout (KO) mice, which were then subjected to DNA ionizing radiation. Irradiation-sensitive tissues such as thymus and spleen were harvested for TUNEL staining. **g** Representative images of TUNEL staining of spleen and thymus from wild type mice or p53 KO mice injected with indicated ARMMs. 2-month old mice (n = 3-4 for each condition) were injected with A1 or A1-p53 ARMMs and 36 h later subjected to irradiation. Age-matched WT animals were used as positive controls for irradiation treatment. Scale bar for the images are shown. **h** Quantification of apoptotic (TUNEL-positive) cells in tissue staining images. Data were presented as mean ± SEM. *$p < 0.05$; **$p < 0.01$

control ARRDC1-GFP. Mice with wild-type p53 were included as a positive control. We irradiated the mice to induce DNA damage and then harvested irradiation-sensitive tissues such as thymus and spleen for TUNEL staining to evaluate apoptosis (Fig. 1f). As expected, irradiation induced significant apoptosis in the spleen and thymus of mice with wild-type p53 (Fig. 1g, top panels; Fig. 1h), whereas p53 KO mice injected with control ARMMs containing ARRDC1-GFP were almost completely resistant to irradiation-induced apoptosis (Fig. 1g middle panels, Fig. 1h). Importantly, in p53 KO mice that received intravenous injection of ARRDC1-p53 ARMMs, there was significant induction of apoptosis post irradiation in both spleen and thymus (Fig. 1g lower panels, Fig. 1h). This result indicates that ARMMs can

efficiently package and deliver functional p53 into multiple tissues in vivo.

**Packaging and delivery of RNAs via ARMMs.** RNA molecules are also broadly used as therapeutic agents[15], but often have to overcome cellular barriers[16]. We tested the ability of ARMMs to package and deliver RNAs to recipient cells. To package RNAs into ARMMs, we took advantage of the transactivator of transcription (Tat) protein, which binds specifically to the stem-loop-containing trans-activating response (TAR) element RNA[17, 18]. We made an expression construct with a short Tat peptide fused directly to the C-terminus of ARRDC1 and another construct

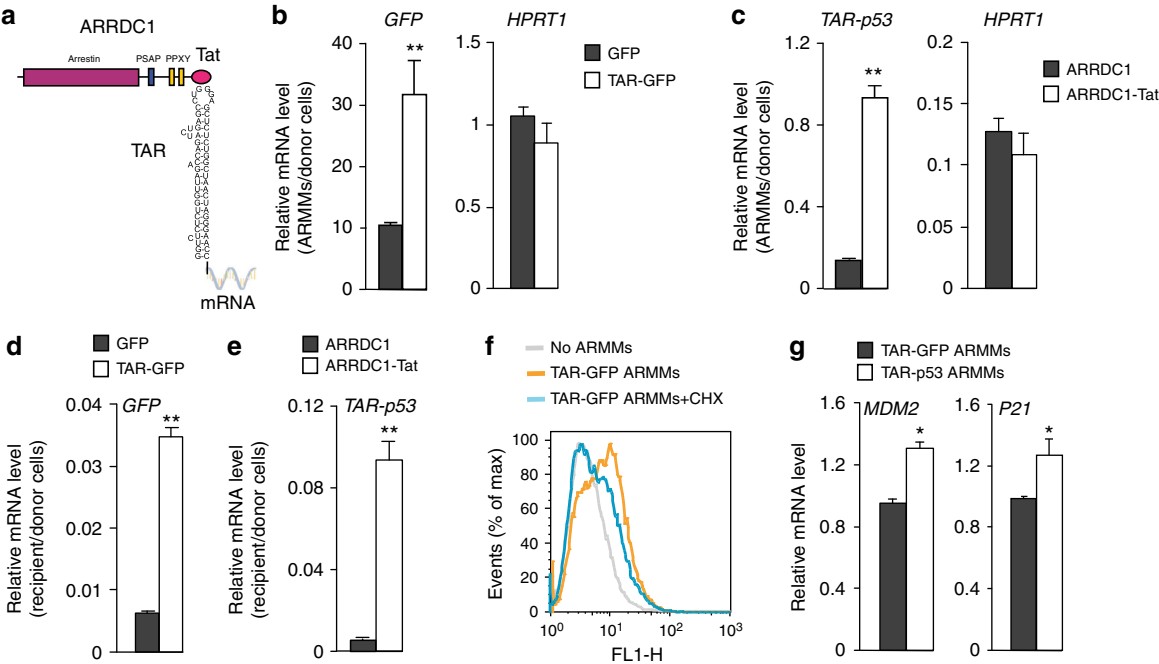

**Fig. 2** Packaging and delivery of RNAs via ARMMs. **a** RNA packaging strategy. TAT peptide, which binds specifically to TAR, is fused to the C-terminus of ARRDC1 to recruit RNA cargo molecules linked to TAR, into ARMMs. **b** Packaging of TAR-GFP mRNA in ARMMs. ARRDC1-TAT was co-transfected with TAR-GFP or control GFP construct into HEK293T cells. ARMMs were pelleted via ultracentrifugation. qRT-PCR was done on ARMMs and on the transfected cells for GFP and for a control mRNA (HPRT1). **c** Packaging of TAR-p53 mRNA in ARMMs. TAR-p53 was co-transfected with ARRDC1 or ARRDC1-TAT construct into HEK293T cells. ARMMs were pelleted via ultracentrifugation. qRT-PCR was done on ARMMs and on the transfected cells for TAR-p53 and for HPRT1. **d** Transfer of TAR-GFP mRNA into recipient cells. A549 cells were incubated with ARMMs containing TAR-GFP mRNA overnight, washed with PBS extensively, and subjected to mRNA analysis by qRT-PCR. **e** Transfer of TAR-p53 mRNA into recipient cells. p53-null H1299 cells were incubated with ARMMs containing TAR-p53 mRNA overnight, washed with PBS extensively, and subjected to mRNA analysis by qRT-PCR. **f** Translation of ARMMs-delivered GFP mRNA in recipient cells. A549 cells were incubated with ARMMs containing TAR-GFP mRNA for 24 h with or without the translational inhibitor cycloheximide (CHX), and subjected to flow cytometry analysis. **g** Activation of p53 target genes in recipient cells receiving TAR-p53 ARMMs. P53-null H1299 cells were incubated with ARMMs containing TAR-p53 mRNA for 18 h and subjected to mRNA analysis by qRT-PCR to detect MDM2 and p21 mRNAs. At least three independent replicates were done for all assays. Data were presented as the mean ± SD in the bar graphs. *$p < 0.05$; **$p < 0.01$

with TAR fused directly to the 5′ end of a cargo mRNA (Fig. 2a). We reasoned that the high binding affinity between the Tat peptide and TAR will allow the recruitment of the TAR-fused mRNA into ARMMs. We tested the packaging efficiency of both GFP and p53 mRNAs into ARMMs. We transfected ARRDC1-Tat with control GFP or TAR-GFP into production cells, and harvested ARMMs for mRNA and protein analysis. GFP mRNAs were significantly more enriched in ARMMs of ARRDC1-Tat and TAR-GFP co-transfection (Fig. 2b). Similarly p53 mRNA fused to TAR was significantly enriched in ARMMs when co-expressed with ARRDC1-Tat (Fig. 2c). No GFP or p53 proteins were detected by Western blot in either GFP or TAR-GFP-mRNA-containing ARMMs (Supplementary Fig. 5), indicating that the Tat-TAR system selectively packaged TAR-labeled mRNAs into ARMMs. We next determined whether the TAR-GFP (or TAR-p53) mRNA in ARMMs can be delivered into and expressed in recipient cells. Incubation of ARMMs containing TAR-fused mRNAs with recipient A549 cells led to the detection of high level of GFP or p53 mRNAs in the recipient cells (Fig. 2d, e). Importantly, flow cytometry analysis confirmed that GFP mRNAs in the recipient cells were translated into GFP proteins and this translation was nearly abolished in the presence of translation inhibitor cycloheximide (CHX) (Fig. 2f). Incubation of ARMMs containing TAR-p53 increased transcription of Mdm2 and p21 in the recipient cells (Fig. 2g), indicating that TAR-p53 mRNAs

delivered via ARMMs were translated into functional p53 proteins.

**Packaging and delivery of CRISPR-Cas9 via ARMMs.** Having shown that ARMMs can package and deliver both proteins and RNAs, we next tested the utility of ARMMs in delivering more complex macromolecules, such as an RNA/protein complex. The CRISPR-Cas9 system uses an RNA-guided DNA nuclease (Cas9) to directly edit genes in the genome[19, 20]. Because of its high efficiency, specificity, and ease of use, CRISPR-Cas9 has been widely used in biomedical research for gene editing and genome engineering. However, the enormous therapeutic potential of CRISPR-Cas9 is limited by the paucity of suitable delivery methods[21, 22]. Therefore, we explored the use of ARMMs in delivering Cas9 and its associated single-guide RNA (sgRNA) into cultured cells. Because the Cas9 protein is relatively large (molecular weight around 160 kDa), we decided not to use direct fusion to ARRDC1. Instead, we took advantage of the interaction of ARRDC1 with WW-domain-containing proteins. Through its PPXY motifs, ARRDC1 specifically interacts with the WW domains (~40 amino acids each) of NEDD4 family proteins[9, 23], allowing the recruitment of proteins into ARMMs[9]. We made a fusion construct in which Cas9 is linked to either two or four WW domains of the ITCH protein (Fig. 3a). WW-Cas9 fusion

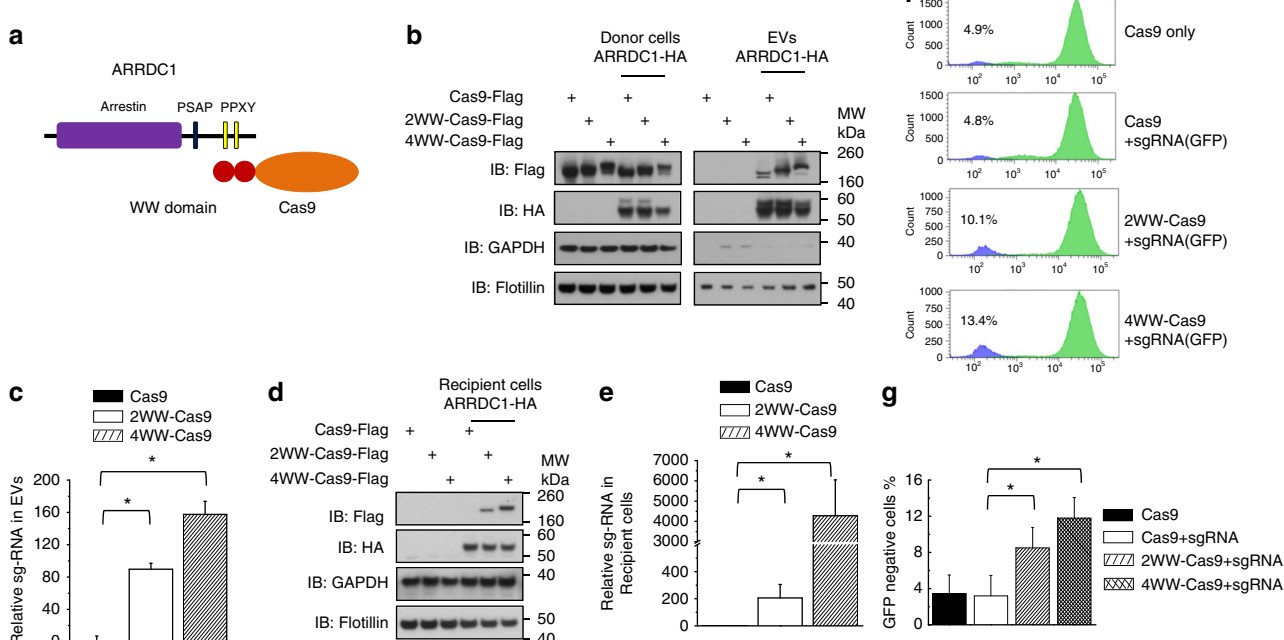

**Fig. 3** Packaging and delivery of CRISPR-Cas9 via ARMMs. **a** Cas9 packaging strategy. Cas9 is fused to WW domains, which interacts with PPXY motifs of ARRDC1, allowing the recruitment of Cas9 into ARMMs. **b** Western blotting showing WW-linked Cas9 in EVs. HEK293T cells were transfected with Cas9, 2WW-Cas9, or 4WW-Cas9 together with pCDNA3.1 or ARRDC1-HA. Extracellular vesicles were pelleted by ultracentrifugation. Both cell lysates and extracellular vesicle (EV) pellets were subjected for anti-Flag, anti-HA, anti GAPDH, and anti-flotillin Western blotting. RNAs were also extracted from EV fractions for qRT-PCR quantification of guide RNA (**c**). **d** Transfer of WW-domain linked Cas9 proteins in ARMMs to recipient cells. HEK293T cells were incubated with different ARMMs overnight, washed with PBS extensively, and subjected to Western blotting. RNAs were also extracted from the recipient cells for qRT-PCR quantification of guide RNA (**e**). **f** Flow cytometry data showing the WW-domain linked Cas9 functioned in the recipient cells, which were U2OS cell with stable expression of GFP protein. **g** Quantifications of flow cytometry data (**f**) from three independent replicates. Data were presented as the mean ± SD in the bar graphs. *$p < 0.05$

proteins showed similar gene editing activity to that of unmodified Cas9 (Supplementary Fig. 6), indicating that WW fusion did not affect the activity of CRISPR-mediated gene editing. WW-Cas9 incorporation into ARMMs was determined by Western blotting. As shown in Fig. 3b, in the absence of ARRDC1, there was very little release of Cas9 or WW-Cas9 in EVs. However, when co-expressed with ARRDC1, WW-domain fusion to Cas9 was robustly detected in ARMMs, while little unmodified Cas9 was found in ARMMs. To further demonstrate the incorporation of WW-Cas9 into ARMMs, we fractionated WW-Cas9 EV pellet using a sucrose gradient that has previously shown to be able to fractionate and isolate ARMMs[10]. As shown in Supplementary Fig. 7, WW-Cas9 co-segregates with ARRDC1, indicating that WW-Cas9 fusion protein is likely incorporated into ARMMs. In addition to WW-Cas9 fusion proteins, sgRNA was also robustly detected into ARMMs (Fig. 3c). These results indicate that WW-Cas9 along with its associated sgRNA was incorporated into ARMMs.

We next determined whether the Cas9/sgRNA packaged into ARMMs can be delivered to recipient cells. We incubated the purified ARMMs with cells containing a single copy of the GFP gene[24]. Both Western blotting (Fig. 3d) and confocal imaging (Supplementary Fig. 8a) showed the presence of WW-Cas9 proteins in recipient cells. In addition, the sgRNA that targets the GFP gene was also robustly detected in the recipient cells (Fig. 3e). These results indicate the delivery of Cas9/sgRNA from ARMMs into the target cells. We then measured gene editing in the cells. Incubation of ARMMs containing WW-Cas9 and anti-GFP sgRNA led to a significant increase of GFP-negative cells (Fig. 3f, g) and disappearance of GFP signal in some cells (Supplementary Fig. 8b), suggesting inactivation of the GFP gene

by Cas9/anti-GFP-sgRNA. T7E1 assay and direct DNA sequencing both confirmed gene editing in the target GFP gene (Supplementary Fig. 8c, d).

## Discussion

In summary, we have shown that ARMMs can be used to efficiently package and deliver diverse classes of cargo macromolecules. Several strategies were employed to package these different cargos: Direct fusion to ARRDC1 to recruit p53 proteins, fusion to WW domains for the large macromolecular CRISPR complex, and a peptide-RNA interaction for mRNA molecules. In all cases, we showed selective packaging of the respective cargos into ARMMs. Importantly, we demonstrate that the cargo molecules packaged in ARMMs, when transferred to cultured recipient cells, carry out the expected biological function within recipient cells. For example, we demonstrate that p53 delivered via ARMMs induce robust DNA damage-dependent apoptosis in multiple tissues in *p53*-null mice. Together, these studies provide the proof-of-principle demonstration for the utility of ARMMs as a versatile platform for packaging and in vivo delivery of macromolecules.

ARMMs offer several advantages over existing delivery methods. For example, the use of transducing peptides[25], chemical induction[26], and a variety of nano-carriers for delivery[27, 28] have had marginal success in vitro and even ex vivo[29], yet, these approaches suffer from myriad problems in vivo, including low specificity, instability, and high immunogenicity[30, 31]. ARMMs are produced endogenously and thus are likely less immunogenic and less toxic. ARMMs are generally stable at physiological conditions with cargo molecules protected from degradation. In

addition, ARMMs may be engineered to present "homing" molecules (e.g., peptides and antibodies) on the surface to allow for tissue/cell-specific delivery. Moreover, because of the similarity of biogenesis between ARMMs and fusogenic viruses[32, 33], there is a possibility that ARMMs may be taken up by cells via membrane fusion, which would allow ARMMs and the packaged cargos to avoid the lysosomal degradation machinery. Future studies exploring these properties of ARMMs will further develop these unique EVs into a versatile and potentially superior platform for intracellular delivery of bio-therapeutics.

## Methods

**Plasmids.** ARRDC1-GFP expression construct was made previously[9]. pcDNA3-p53 expression construct was made by cloning full-length DNA fragment of wild-type p53 into the pcDNA3.1(+) vector (between the EcoRI and XhoI sites). ARRDC1-p53 fusion construct was made by cloning the full-length *ARRDC1* DNA into pcDNA3-p53 (between the NheI and EcoRI sites) upstream of the *p53* gene. New constructs were confirmed by direct DNA sequencing. To generate ARRDC1-TAT construct, the DNA sequence of *ARRDC1* was PCR amplified followed by insertion into pcDNA3 vector to obtain pcDNA3-ARRDC1 construct. The DNA sequence of TAT (48–65 aa) was synthesized, annealed, and inserted at the C-terminus of ARRDC1. The DNA sequence of TAR (1–63 base pairs) was synthesized, annealed, and inserted at the 5′ end of EGFP in the pEGFP-N1 vector (Addgene) to obtain the TAR-EGFP construct. Alternatively, the TAR region was inserted at the 5′ end of p53 in the pcDNA3-*p53* construct to obtain the TAR-p53 construct. 2WW-Cas9-anti-GFP and 4WW-Cas9-anti-GFP were constructed by cloning either two or four WW domains from ITCH into the AgeI site of the px-330 vector (Addgene).

**Mammalian cell culture and transfections.** HEK293T, A549, and H1299 cells were obtained from ATCC. HEK293T and A549 cells were cultured in DMEM (high glucose) (Gibco), supplemented with 10% fetal bovine serum (FBS) (Gibco), and 100 μg/ml PenStrep (Gibco). H1299 cells were cultured in RPMI medium (Gibco), supplemented with 10% FBS and 100 μg/ml PenStrep. Cells were grown at 37 °C in 5% $CO_2$. All cell cultures were regularly checked for mycoplasma contamination. Transfections in HEK293T and H1299 cells were performed using Fugene 6 (Promega) and TurboFect (Thermo Fisher Scientific), respectively, according to the manufacturers' instructions. For the CHX treatment, A549 cells were incubated with ARMMs for 3 h, and 2 μg/ml CHX were added to incubate for additional 24 h, followed by the flow cytometry analysis.

**Nanoparticle tracking analysis (NTA).** EVs or ARMMs were analyzed and quantitated by the NanoSight LM10 instrument with the NTA software (Malvern). Samples containing vesicles were diluted with phosphate-buffered saline (PBS) and recorded for 60 s with five repeats. Videos were analyzed by the NTA software to obtain an average of the five repeats for each sample.

**ARMMs purification and incubation.** Briefly, HEK293T cells were transfected with different constructs as indicated. Six hours later, the medium was replaced with fresh DMEM supplemented with 10% pre-cleared FBS, which was subjected to ultracentrifugation at $260,800 \times g$ for 4 h to remove any bovine vesicles. Seventy-two hours after transfection, the cell culture supernatant was collected and subjected to two consecutive rounds of centrifugation ($500 \times g$ and $2000 \times g$). The medium was then passed through a 200 nm filter (Acrodisc) and subjected to ultracentrifugation using the SW41Ti rotor in a L8-M or XE90 centrifuge (Beckman) at $166,900 \times g$ for 2 h. the medium was then aspirated, and the pellets enriched with ARMMs were washed twice with ice-old PBS. The vesicles were then resuspended in RPMI medium supplemented with 10% pre-cleared FBS for incubation with recipient cells. Alternatively, the vesicles were resuspended in PBS for animal injection. The purity and yield of vesicles were measured using NanoSight NS300 instrument (Malvern).

**Sucrose gradient fractionation of ARMMs.** ARMMs/EV pellets were further fractionated by the sucrose gradient method as described[10]. Briefly, EV pellets were washed twice, resuspended in PBS and reloaded onto a sucrose gradient of ten different sucrose concentrations from top to bottom (0.2–2 M) and centrifuged at $180,000 \times g$ for 18 h. Fractions were then carefully collected at 1 ml each from the bottom of the tube. All fractionated samples were diluted with PBS and subjected to centrifugation at $180,000 \times g$ for 90 min to pellet the vesicles.

**Negative staining TEM.** EVs isolated by the sucrose gradient purification procedure were suspended in 20 μl of PBS, vortexed briefly, and adsorbed for 1 min to a carbon coated grid that had been made of hydrophilic by a 30 s exposure to a glow discharge. Excess liquid was removed with filter paper and the samples were stained with 0.75% uranyl formate for 30 s. After removing the excess uranyl

formate the grids were examined in a JEOL 1200EX TEM and images were recorded with an AMT 2k CCD camera.

**ARMMs transfer in transwell assay.** Transfected cells were washed thoroughly and seeded atop a 0.4-μm transwell membrane (Costar) for 24 h. The transwells were then transferred to a plate containing untransfected HEK293T cells. ARMMs transfer was allowed to proceed for 30 h before harvesting. Alternatively, purified ARMMs resuspended in PBS were added to culture medium containing untransfected HEK293T cells and were incubated for 24–30 h before harvesting.

**Immunoblot analysis and antibodies.** Cells were lysed in NP-40 lysis buffer (0.5% NP-40, 50 mM Tris-HCl, and 150 mM NaCl) supplemented with a protease inhibitor mixture (Roche). Lysates or vesicles resuspended in lithium dodecyl sulfate sampling buffer (Novex) were resolved on a 4–12% NuPAGE gel (Novex) and transferred onto a PVDF membrane (Bio-Rad). Blots were probed with primary antibodies in Tris-buffered saline containing 0.1% Tween 20 and 5% Nonfat milk, followed by HRP-conjugated anti-rabbit antibody (Cell Signaling, 7074S, at 1:3000 dilution) or anti-mouse antibodies (Cell Signaling, 7076S, at 1:3000 dilution). Primary antibodies used include anti-ARRDC1 rabbit polyclonal antibody (in house, ref.[9], 1:2000), anti-vinculin rabbit monoclonal antibody (Abcam, AB129002, at 1:1000 dilution), anti-p53 mouse monoclonal antibody (Santa Cruz, SC-126, 1:250), anti-GAPDH rabbit polyclonal antibody (Santa Cruz, SC-25778, 1:250), rabbit monoclonal GFP antibody (Cell Signaling, 2956S, 1:1000), monoclonal FLAG antibody (Sigma M2, F1804, 1:2000), rabbit monoclonal HA-Tag antibody (Cell Signaling, 3724S, 12117S, 1:2000), and rabbit monoclonal Flotillin antibody (Cell Signaling, 18634S, 1:2000). Uncropped scans of Western blots were included in Supplementary Fig. 9.

**Quantitative RT-PCR.** Total RNAs were extracted from cells or ARMMs using RNeasy mini kit (Qiagen) and subjected to the treatment of DNase I (Invitrogen). First strand cDNA synthesis was done using SuperScript III for RT-PCR (Invitrogen) per the manufacturer's instructions. A StepOnePlus real time PCR system (Applied Biosystems) was used for quantitative PCR analysis of cDNAs.

**Immunofluorescence staining and confocal imaging.** Cells grown on glass coverslips were washed with PBS thrice and then incubated in 4% paraformaldehyde (Sigma) for 20 min on ice. Cells were permeabilized using 0.1% Triton X-100 in PBS for 5 min on ice and then blocked in the blocking buffer (2% Bovine serum albumin in PBS) for 1 h at room temperature. The cells were incubated with primary antibody anti-p53 mouse monoclonal antibody (Santa Cruz, SC-126, 1:200), followed by secondary antibody anti-mouse-alexa fluor 647 (Invitrogen, A21236, 1:500). The glass coverslips were mounted on glass slides using Prolong Gold Antifade Mountant with DAPI (Thermo Fisher Scientific, P36941). Image acquisition was carried out using a Leica TCN-NT laser-scanning confocal microscope (Leica) equipped with air-cooled argon and krypton lasers. Images were processed using ImageJ.

**Flow cytometry.** Cells were trypsined from the plate and washed with PBS, then labeled with 7-aminoactinomycin D to gate away the dead cells. Flow cytometry was done using BD FACSAria color sorter. Data were analyzed using FlowJo.

**Mouse studies.** TP53 KO mice (B6.129S2-Trp53tm1Tyj/J) were purchased from the Jackson Laboratories. Healthy 2-month old mice without obvious signs of tumor development were injected intravenously with ARMMs ($6 \times 10^{10}$ EVs/mouse) in sterile PBS. Thirty-six hours later, mice were irradiated (4 Gy) using a Cs-137 irradiator. Thymus and spleen were collected 3 h after irradiation and fixed in 10% phosphate-buffered formalin. All mouse studies were carried out according to guidelines approved by the Institutional Animal Care and Use Committee of the University of Massachusetts Medical School.

**TUNEL staining.** Thymus and spleen fixed in 10% formalin were paraffin-embedded. Five-micrometer sections were stained for TUNEL using an in situ cell death detection kit (Roche) by the University of Massachusetts Medical School Morphology Core facility. For each sample, 3–5 immunohistochemistry images were obtained using an Axiovert 40 CFL microscope (Carl Zeiss) with a CCD camera (QImaging QI Click), and QCapture Pro 7 software (QImaging). The images were analyzed using ImageJ software. Each image was color channel-split and then thresholded to calculate the percentage of TUNEL-positive area.

**Statistics.** Statistics were calculated using Excel. Comparisons of two groups were analyzed using two-tailed *t* test as indicated. Data were presented as the mean ± standard deviation (SD) or standard error of the mean (SEM). *P* values <0.05 were considered significant, as indicated by *. *P* < 0.01 was indicated by **. All the cell culture data shown in the manuscript are representative of experiments conducted at least thrice.

**Data availability**. The data that support the findings of this study are available from the corresponding author upon request.

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

## Acknowledgements

We thank Dr. Grant Zimmermann for insightful discussions and suggestions. This work was supported by the Blavatnik Biomedical Accelerator Fund (to Q.L.). Q.L. is also supported in part by the National Institutes of Health grants (R01HL114769, R01ES022230 and P30ES000002) and by the American Asthma Foundation Scholar award (AAF15–0097).

## Author contributions

Q.L. conceived and supervised the project. Q.W., J.Y., Z.H. and Q.L. designed the experiments. Q.W., J.Y., and T.K. performed the experiments with the help of J.B. Q.W., J.Y., T.K., Z.H. and Q.L. analyzed the data. Q.W., J.Y., Z.H. and Q.L. wrote the manuscript with the help of all coauthors.

## Additional information

**Competing interests:** Harvard University has filed various patent applications regarding the ARMMs technology on behalf of the inventors with Q.L. and Q.W. listed as co-inventors. The remaining authors declare no competing interests.

