## [Peer Review File · Nature Communications]

Reviewers' comments:

Reviewer #1 (Remarks to the Author):

This is an exciting paper discussing a new approach to intracellular delivery . The basis of the paper is the use of the ARMMs system of EV formation and the purpose of the paper is to show the versatility of the system in delivering fusion proteins, mRNA and finally complex systems (CRISPR/cas9) .

The paper is concise and though the data are of a preliminary nature the results are sufficient to inspire many to try the system. The most obvious problem with the paper is the lack of quantiation which makes it hard to judge the efficiency of the system. While this is not an overwhelming criticism the delivery of a molecule which could be precisely quantified would be enormously helpful.

This could in fact be achieved using GFP but measuring it precisely using standardized western blots and or fluorecence methods. How many molecules gat into how many cells from what number of EVs?

Reviewer #2 (Remarks to the Author):

The manuscript by Yu et al. examines whether small extracellular vesicles called ARMMs can be used for therapeutic purposes. The driving idea is that ARRDC1, a scaffold protein that is ubiquitinated and can drive formation of these vesicles by recruiting TSG101 to the cell surface, can be exploited to deliver proteins and RNAs via ARMMs. This is tested using 1) p53 fused to ARRDC1; 2) leveraging the TAT-TAR binding system. mRNAs were fused to a stem-loop sequence (TAR) that binds to a peptide TAT, which was fused to ARRDC1. Both GFP mRNA and p53 mRNA were fused to TAR and delivered via ARRDC1-TAT fusions; and 3) Cas9-guide RNA delivery. In this case, the Cas9 was fused to WW domains, which bind ARRDC1. To test delivery of p53 protein, a very nice in vivo test of radiation sensitivity was used, in which ARMMs containing ARRDC1-p53 fusions were injected in vivo and tested for differences in radiation apoptotic response in the spleen and thymus. To test delivery of p53 and GFP RNA with the TAR-TAT system, both PCR of recipient cells and biological responses (levels of p53 target genes or expression of GFP protein) were tested and observed. For the WW-Cas9 delivery, gene editing of a chromosomal copy of GFP gene in U2OS cells was used. In all cases, the specific targeting of protein or RNAs via ARMMs was effective. Thus, this manuscript establishes a proof of principle for leveraging the targeting and EV-inducing ability of ARRDC1 for protein and/or RNA delivery. This could be an important step for the field – to have such tools to manipulate and deliver proteins and RNAs via EVs. Overall the experimental design is good and the experiments are rigorously executed. Some minor things could be done to improve the paper:

1. The vesicle Western blots should include some vesicle markers – flotillin, TSG101, etc.
2. Some electron microscopy images of the vesicles would be good, to meet typical EV standards of characterization, with a wide field shown.
3. Fig 1, analysis of TUNEL staining from mouse tissues – looks like it was done manually. Should be complemented by an automated method that can use the whole image (for example, TUNEL-positive area per image area could be done from thresholded images). Or was the TUNEL-positive cells done automatically?
4. Fig 1, some better labeling of the figure could be done – does donors mean donor cells? Likewise, recipients = recipient cells?
5. Fig 3B and 3D, likewise, the panels could benefit from some explicit titles across the top so that the reader doesn't have to hunt in the figure legend to figure out what it is.
6. Fig 3, as another test of the ability of WW-Cas9 to clip out GFP, it would be good to show IF images of GFP in cells to complement the FACS analysis.
7. Fig S3, why do the ARRDC1 bands in the vesicles look different between panels a and b? Was

the gel cropped closer in a?

We thank the reviewers for their helpful and constructive comments. Changes made to the manuscript are highlighted (vertical line to the left of revised paragraphs, or **boxed** for text). Below I detail our responses to each of the critiques. For clarity, reviewer comments are italicized and followed by our specific responses.

Response to comments by reviewer #4

The most obvious problem with the paper is the lack of quantiation which makes it hard to judge the efficiency of the system. While this is not an overwhelming criticism the delivery of a molecule which could be precisely quantified would be enormously helpful. This could in fact be achieved using GFP but measuring it precisely using standardized western blots and or fluorecence methods. How many molecules gat into how many cells from what number of EVs?

Response: We have obtained standard recombinant GFP protein (Novus, NBC1-22949) as suggested and used it in Western blotting to quantify the efficiency of ARMMs packaging and transfer. To quantify ARMMs packaging, we ran the GFP standard at a range of 50 to 1000 ng along with sample of ARMMs produced from ARRDC1-GFP-expressing cells (Supplementary Fig. 3a). From the standardized plot we estimated that each ARMM vesicle contains an average of ~540 cargo protein molecules (ARRDC1-GFP) (Supplementary Fig. 3a). To quantify the amount of ARMMs transfer to recipient cells, we used GFP standard at lower concentrations as only a fraction of ARRDC1-GFP in ARMMs got transferred to the recipient cells. We estimated that each recipient cell received an average of 3.1×10^6 cargo protein molecules from 5.8×10^3 of ARMMs (Supplementary Fig. 3b). These new data provide us with a ball park estimation of the efficiency of ARMMs packaging and transfer. We acknowledge that future studies are needed to quantify the in vivo efficiency and to compare ARMMs directly with other delivery platforms.

Response to comments by reviewer #2

Some minor things could be done to improve the paper:

1. The vesicle Western blots should include some vesicle markers – flotillin, TSG101, etc.

Response: We have added flotillin Western blotting to Fig 1c,d, and Fig 3b,d.

2. Some electron microscopy images of the vesicles would be good, to meet typical EV standards of characterization, with a wide field shown.

Response: We have used transmission electron microscopy (TEM) to image EVs isolated from cells expressing either control GFP, ARRDC1-GFP, ARRDC1-p53, or ARRDC1-HA (co-transfected with 4WW-Cas9). The diameter of most EVs is less than 100 nm, consistent with our previous calculation of 50-80 nm (Nabhan 2012 PNAS). While not particularly quantitative, the TEM data are consistent with our NanoSight data showing

more EVs in ARRDC1-overexpressing samples than in the GFP control. The TEM data are included as Supplementary Fig. 2.

3. Fig 1, analysis of TUNEL staining from mouse tissues – looks like it was done manually. Should be complemented by an automated method that can use the whole image (for example, TUNEL-positive area per image area could be done from thresholded images). Or was the TUNEL-positive cells done automatically?

Response: We have reanalyzed the TUNEL staining data using the ImageJ software. The new data showed similar trends: in *p53* KO mice that received intravenous injection of ARRDC1-p53 ARMMs, there was significant induction of apoptosis post irradiation in both spleen and thymus. The old data (done by manual counting) is now replaced with the new data (automated analysis) (Fig. 1h).

4. Fig 1, some better labeling of the figure could be done – does donors mean donor cells? Likewise, recipients = recipient cells?

Response: We have modified the labeling of the panels as “Donor cells” or “Recipient cells”.

5. Fig 3B and 3D, likewise, the panels could benefit from some explicit titles across the top so that the reader doesn't have to hunt in the figure legend to figure out what it is.

Response: We have added new labeling to clearly mark the panels.

6. Fig 3, as another test of the ability of WW-Cas9 to clip out GFP, it would be good to show IF images of GFP in cells to complement the FACS analysis.

Response: We have included the fluorescence images of GFP in cells (Supplementary Fig. 7b). Our data clearly showed GFP-negative cells in samples that received ARMMs containing Cas9.

7. Fig S3, why do the ARRDC1 bands in the vesicles look different between panels a and b? Was the gel cropped closer in a?

Response: Fig S3 is now the new Supplementary Fig. 5. The upper band of ARRDC1 blot in panel a was cropped out. We have replaced the blot with a full gel picture of ARRDC1. For all gel pictures in this manuscript we have included the scans of the original films as Supplementary Figure 8.

Reviewers' comments:

Reviewer #1 (Remarks to the Author):

I was very pleased to see the response to the comments . The very good quantitation data really help this paper .The Number of molecules of p53 that can be delivered supports the biological data strongly and creates a great deal more confidence in the results. the additional data asked for by the other reviewer has also been provided so i am happy with this version of the manuscript . The system has great potential to join the tool kit of delivery methods .

Reviewer #2 (Remarks to the Author):

Most of my concerns have been addressed. However, there are two remaining points:

1. The new TEM images in Supplementary Fig 2 are a bit troubling because they suggest that the 4WW-Cas9+ARRDC1-HA preparation has extra contamination with a small particle which could be extravesicular protein aggregates or high density lipoprotein that could be carrying the engineered cargo. ImmunoEM to show that the Cas9 is inside of the EVs would allay this concern. Or doing an extra purification with density gradient to show that the Cas9 segregates with the EVs and not with protein aggregates.
2. There is a notable lack of molecular weight markers on the Western blots in the supplement.

We thank the reviewers for their helpful and constructive comments. Changes made to the manuscript are highlighted (vertical line to the left of revised paragraphs, or boxed for text). Below I detail our responses to each of the critiques. For clarity, reviewer comments are italicized and followed by our specific responses.

Response to comments by reviewer #1

I was very pleased to see the response to the comments. The very good quantitation data really help this paper. The Number of molecules of p53 that can be delivered supports the biological data strongly and creates a great deal more confidence in the results. the additional data asked for by the other reviewer has also been provided so i am happy with this version of the manuscript. The system has great potential to join the tool kit of delivery methods.

Response: Thank you.

Response to comments by reviewer #2

1. The new TEM images in Supplementary Fig 2 are a bit troubling because they suggest that the 4WW-Cas9+ARRDC1-HA preparation has extra contamination with a small particle which could be extravesicular protein aggregates or high density lipoprotein that could be carrying the engineered cargo. ImmunoEM to show that the Cas9 is inside of the EVs would allay this concern. Or doing an extra purification with density gradient to show that the Cas9 segregates with the EVs and not with protein aggregates.

Response: To demonstrate that WW-Cas9 is incorporated into ARMMs/EVs but not associated with smaller protein aggregates, we performed density gradient purification as the reviewer suggested. We used the sucrose gradient procedure that we previously shown to be able to fractionate and isolate ARMMs (Wang and Lu, Nat. Commun. 2017, ref. 10). We fractionated the EV pellet (collected from HEK293T cells co-transfected with WW-Cas9-Flag and ARRDC1-HA) onto the sucrose gradient (0.2 -2 M) and collected 10 fractions, which were further spun down by ultracentrifugation. We did Western blotting on the 10 fractions for HA (to detect ARMMs) and FLAG (to detect WW-Cas9, which is tagged with FLAG). As shown in new **Supplementary Fig. 7**, WW-Cas9 signal co-segregates very well with ARRDC1. This new data indicate that WW-Cas9 co-segregates with ARMMs and allay the concern that WW-Cas9 may be associated with smaller protein aggregates. A brief description of sucrose gradient procedure is added to the Methods section.

2. There is a notable lack of molecular weight markers on the Western blots in the supplement.

Response: We have added the molecular weight markers to all Western blots in the Supplementary Figures (**Supp Fig. 1, 3, 5 and 7**) as suggested.